# Economic Evaluation of Proactive PTSI Mitigation Programs for Public Safety Personnel and Frontline Healthcare Professionals: A Systematic Review and Meta-Analysis

**DOI:** 10.3390/ijerph22050809

**Published:** 2025-05-21

**Authors:** Hadiseh Azadehyaei, Yue Zhang, Yan Song, Tania Gottschalk, Gregory S. Anderson

**Affiliations:** Faculty of Science, Thompson Rivers University, Kamloops, BC V2C 0C8, Canada

**Keywords:** public safety personnel, healthcare professionals, resilience training, return on investment, economic evaluation

## Abstract

Public safety personnel and frontline healthcare professionals are at increased risk of exposure to potentially psychologically traumatic events (PPTEs) and developing post-traumatic stress injuries (PTSIs, e.g., depression, anxiety) by the nature of their work. PTSI is also connected to increased absenteeism, suicidality, and performance decrements, which compromise occupational and public health and safety in trauma-exposed workers. There is limited evidence on the cost effectiveness of proactive “prevention” programs aimed at reducing the risk of PTSIs. The purpose of this meta-analysis is to measure the economic effectiveness of proactive PTSI mitigation programs among occupational groups exposed to frequent occupational PPTEs, focusing on the outcomes related to PTSI symptoms, absenteeism, and psychological wellness. Findings from 15 included studies demonstrate that proactive interventions can yield substantial economic and health benefits, with Return On Investment (ROI) values ranging widely from −20% to 3560%. Shorter interventions (≤6 months) often produced higher returns, while longer interventions (>12 months) showed more moderate or negative returns. Notably, the level at which an intervention is targeted significantly affects outcomes—programs aimed at managers, such as the 4 h RESPECT training course, demonstrated a high ROI and broad organizational impact by enhancing leadership support for employee mental health. Sensitivity analyses highlighted significant variability based on the organizational context, program design, and participant characteristics. The majority of proactive interventions successfully reduced psychological distress and enhanced workplace outcomes, although thoughtful consideration of program design and implementation context is essential.

## 1. Introduction

Frontline healthcare professionals (FHP) and public safety personnel (PSP) play critical roles in maintaining public safety and societal well-being. However, they operate in high-risk environments with frequent exposure to potentially psychologically traumatic events (PPTEs)—exposure to direct or indirect experiences of actual or threatened sexual violence, serious injury, or death [1,2]. Such events place personnel at risk for developing post-traumatic stress injuries (PTSIs) with elevated rates of anxiety, depression, substance use, absenteeism, and turnover, ultimately compromising both individual well-being and organizational performance [1,3]. Their well-being is closely connected to their psychological resilience and the level of support they receive within their organizations and social support networks, which highlights the urgent need for proactive mental health interventions. Moreover, the demanding nature of their work makes them particularly vulnerable to the long-term psychological impacts of PPTE exposure, affecting both their mental health and job performance.

Significant efforts have been made to treat PTSIs through both reactive and proactive prevention programs, which aim to mitigate the condition before symptoms become severe using various approaches. Interventions such as resilience training [3,4], peer support networks [5], crisis intervention teams [6], and stress management workshops [7] have been implemented to reduce the incidence of PTSIs and promote supportive workplace environments. These efforts aim to improve mental health outcomes by both mitigating factors related to PTSIs and fostering a culture of support [3].

Despite the increasing awareness of the importance of mitigating psychosocial risk factors on worker well-being, employers often lack robust economic evaluations to inform their investment decisions [8]. In resource-constrained sectors such as healthcare and public safety, the value of psychological interventions requires careful evaluation. For instance, the Resilience in Stressful Events program (RISE) provides timely, peer-delivered support to reduce post-traumatic risks like absenteeism and job resignation. However, its impact requires formal economic evaluation to justify the investment [4].

Cost-effective analysis is crucial for justifying the investments in health programs within the public sector, where resource allocations compete with service delivery to the public [9]. It is particularly crucial in high-risk occupations like firefighting, where work-related injuries and illness impose a substantial economic burden, exceeding USD 250 billion annually in the U.S., with injuries accounting for over 90% of the costs [10]. The evaluation of the program [10] has demonstrated improvements in firefighter health and well-being, providing valuable insights for decision-makers and the delivery of similar programs. Economic evaluations help determine program efficiency and are critical for informing health policy, program development, and budget decisions [11].

Economic assessments often follow frameworks like the National Institute for Health and Care Excellence (NICE) [12], which consider costs and measure effectiveness in quality-adjusted life years (QALYs). These assessments examine the extent of the benefits gained from various interventions in terms of a participant’s quality of life and survival [12]. However, such evaluations should extend beyond individual psychological outcomes to include organizational considerations, including implementation feasibility, long-term effectiveness, and overall economic viability.

There is emerging evidence of evaluations considering broader organizational outcomes. For example, nurse turnover has been estimated at up to USD 300,000 for every 1% increase [4], emphasizing the financial importance of interventions that support retention. Similarly, manager training initiatives such as the RESPECT mental health program have demonstrated significant occupational benefits, with a Return On Investment (ROI) value of GBP 9.98 for every GBP 1 spent, primarily due to reductions in work-related sick leave [7]. A previous study [8] sought to quantify the ROI of job-related interventions developed to improve employee mental health and productivity within Australian workplaces by systematically analyzing ROI across various job-focused strategies, including redesigning roles, improving job control, and enhancing job security. Their study provides a data-driven framework to support more informed and cost-effective workplace mental health initiatives.

Previous studies have focused on evaluating program outcomes through a variety of economic measures such as ROI, cost-effectiveness ratio (CER), incremental cost-effectiveness ratio (ICER), and net monetary benefit (NMB). However, the lack of standardized assessment methods and the diverse nature of intervention designs contribute to inconsistent findings, particularly in the context of our target population.

The purpose of this meta-analysis is to systematically evaluate the effectiveness of proactive PTSI mitigation programs and provide insights into the economic implications of interventions through synthesizing the cost-effectiveness measures from relevant studies. This review also provides evidence-based recommendations for policymakers, organizational leaders, and mental health practitioners seeking to enhance the resilience and well-being of public safety personnel and frontline healthcare professionals.

## 2. Materials and Methods

A preliminary review of the literature identified a lack of comprehensive data evaluating the effectiveness of proactive PTSI mitigation programs among public safety personnel and frontline healthcare professionals exposed to PPTE, which underlines a significant gap in the research. Meta-analyses are essential for synthesizing evidence and quantifying the impact of interventions, specifically in high-risk occupational settings where psychological wellness is critical. This meta-analysis aims to assess the effectiveness of proactive PTSI mitigation programs on financial outcomes, to provide evidence-based insights for policy and practice.

### 2.1. Study Design

This meta-analysis follows the PRISMA guidelines [13] as illustrated in the PRISMA checklist to ensure a systematic and transparent approach to study selection and data synthesis. The study aims to evaluate the effectiveness of proactive PTSI mitigation programs among public safety personnel and frontline healthcare professionals.

### 2.2. Informational Sources

A comprehensive literature search was conducted across multiple bibliographic databases to ensure the inclusion of relevant interdisciplinary research. Initial exploratory searches in Medline, CINAHL, PsycInfo, and Business Source Complete highlighted the necessity of broader coverage. As a result, SCOPUS and Web of Science were selected as primary databases since both extensively cover the essential health, social sciences, and economic journals. Additionally, the 2 bibliographic database searches were supplemented with 13 studies identified by reviewers through their awareness of the literature. Lastly, 13 papers originally selected for extraction were searched in SCOPUS to identify papers citing them. These were filtered based on previously outlined criteria. This resulted in an additional 294 papers being added for screening.

### 2.3. Search Strategy

A variety of terminology was used (see Table 1) to explore the population of public safety personnel and frontline health professionals (e.g., nurse* OR firefighter* OR police* OR “law enforcement” OR dispatcher* OR paramedic* OR EMT OR “emergency medical technicians” OR “emergency medical services” OR “first responders” OR “correctional officers” OR “emergency personnel”, etc.). These were combined with concepts related to the intervention of resilience training (e.g., stress OR coping OR mindfulness OR hardiness OR “post traumatic” OR traumatic OR resilience* OR “mental health” OR psychoeducation* OR “resiliency training” OR “occupational stress” OR burnout). Finally, these were combined with concepts related to the outcome of “return on investment” (e.g., cost* OR economic* OR “cost benefit” OR “economic evaluation” OR “return on investment” OR ROI.). Results in both databases were filtered to the period 2014–2024 for publication years, review article or article for document types, and English or French languages. Collectivist countries were excluded (e.g., India OR People’s Republic of China OR Iran, etc.) because approaches to public safety personnel are considerably different in North America or the European Union contexts. Results from both databases (1741 from Web of Science and 2063 from SCOPUS and 1 from citation searching) were added to the Covidence systematic review management tool for further analysis by reviewers. 

The target population for this study is public safety personnel and frontline healthcare professionals who often operate in high-demand, low-control settings that require rapid decision making with potentially serious outcomes. These roles are typically structured within strict hierarchies and involve unique occupational demands not commonly seen in other occupations, contributing to increased risk for PTSI and other mental health disorders [2].

Cultural context plays a significant role in shaping psychological safety, with individuals from collectivist cultures often emphasizing group harmony, power distance, and respect for authority. This contrasts with individualistic societies, where personal autonomy and individual expression are prioritized, influencing the dynamics of psychological safety differently. Collectivist societies are characterized by a strong emphasis on interdependence and group-oriented decision making, in contrast with individualistic cultures that prioritize self-reliance and personal goals [14]. Given the contextual specificity of team psychological safety and potential interventions, this review excluded studies originating from collectivist cultural settings.

### 2.4. Selection of Studies

A population-intervention-comparison-outcome (PICO) framework was used to define study variables of interest, and the keywords entered into our systematic literature searches, which are provided in Table 1. We searched across relevant bibliographic databases as mentioned in Section 2.2. The database-facilitated searches were supplemented with additional studies with hand-search reference lists from included studies, as well as previous review articles and reports. Following the searches, all citations were imported into Covidence—a web-based systematic review manager. Two independent reviewers screened articles against the eligibility criteria: first by title/abstract and then in full. Initial screening was verified by having multiple reviewers screen 71 papers, resulting in 100% agreement. All discrepancies were resolved by consensus between the two reviewers.

Data extraction was conducted independently by a single reviewer from full-text reports of eligible articles as per the PICO framework (Table 2). Following a structured framework, study identification variables included the title, authors, and the year of publication. Population characteristics captured the sample size and study design. Intervention-specific details encompassed the description, frequency, duration, and setting. Evaluation methods and financial metrics, such as intervention costs, cost savings, productivity gains, and ROI, were recorded to assess economic impact. Mental health and well-being outcomes, including changes in depression, anxiety, PTSD, burnout, stress, and resilience scores, were also extracted.

In addition to our systematic process of screening, we also identified 14 papers and included them at the full-text review stage. These studies were not covered in the title and abstract screening but were found to be relevant based on their alignment with our inclusion criteria. We also reviewed their references and citations to identify further relevant literature, which helped capture additional high-quality papers that might have been missed in the initial searches.

The systematic review identified a total of 3805 studies. Among the identified studies, there were 753 removed as duplicates, leaving 3052 studies for the title and abstract screening. There were 2967 records removed, leaving 85 studies for full-text review. There were 70 studies excluded at the full-text stage: 8 had an unsuitable population (i.e., not a PSP, FHP, or a PPTE-exposed occupational group), 14 had an unsuitable study design (i.e., not a pre-post evaluation of outcomes, such as qualitative studies or protocols, or non-peer-reviewed dissertations, books, or reports), 3 had an unsuitable intervention (e.g., post-PPTE service, treatment, or therapeutic intervention), and 45 papers did not report an ROI outcome. The systematic review process resulted in 15 eligible studies that evaluated the effectiveness of a proactive PTSI mitigation program in workers exposed to PPTEs. Key study characteristics are described below and are summarized in Table 3, including participant summaries and study designs. Ultimately, 15 studies were included in a quantitative meta-analysis (Figure 1).

### 2.5. Inclusion and Exclusion Criteria

The current review was restricted to peer-reviewed English-language studies assessing the return on investment of mental health programs designed to proactively mitigate the impact of PPTEs among adult (aged 18 and older) public safety and front-line healthcare workers and published since 1 January 2015. Eligible study designs included those assessing pre- and post-intervention costs with an analysis of return on investment. Studies involving participants with one or more identifiable mental disorders (e.g., clinician diagnosis or a positive screen on a validated psychological instrument), non-PPTE occupational stressors (e.g., work-related demands, organizational stress), or non-experimental designs (e.g., protocols, theses, qualitative studies) were excluded.

## 3. Results

### 3.1. Study Selection and Characteristics

We included 15 studies after screening 3052 interventions related to public safety personnel and frontline healthcare professionals who are exposed to potentially psychologically traumatic events across different sectors, such as healthcare, police, firefighters, and dispatchers. These studies were primarily based in the USA, UK, Netherlands, Australia, and Germany.

### 3.2. Research Design and Methodology

The research designs varied in evaluating the economic impact and effectiveness of resilience interventions. Several studies used randomized controlled trials (RCTs) [7,9,16,17,18,19], such as intent-to-treat (ITT) analysis and pragmatic effectiveness trials [19]. Economic evaluations, such as cost-effectiveness analysis (CEA) and cost–consequence analysis [20], were applied to compare scenarios like baseline burnout prevalence versus intervention outcomes. Markov models with a one-year time horizon were applied in some studies [4,5]. Furthermore, there were also observational studies [22], such as retrospective cohort designs [23], and quasi-experimental analyses [10], which were part of these studies, assessing real-world intervention effects.

Sample sizes in the studies ranged widely from 80 to 1454 participants, while some studies applied hypothetical cohorts. Follow-up rates also varied across different studies, which this change affecting cost measures. While smaller sample studies introduced variability and impact the overall estimates, larger samples enhanced statistical power. Table 3 presents a summary of the key characteristics of the studies included in this analysis. The table provides essential details, including study quality, sample size, population demographics, study design, program descriptions, duration, evaluation methods, key outcomes, and cost-saving results.

### 3.3. Intervention Types and Objectives

Most interventions were designed with proactive and preventive elements, emphasizing the early identification of mental health risks (e.g., screening-based interventions with occupational physician referrals) or resilience-building (e.g., peer-support programs for second victim support). However, the interventions addressed different aspects of resilience training, targeting different components of mental well-being. Prevention-focused programs, such as burnout reduction programs [19] and workplace wellness programs like PHLAME [10], aimed to foster long-term well-being. Resilience was reinforced through structured emotional and psychological assistance offered by programs like the Peer Support Program [5] and the Crisis Intervention Team [6]. Digital mindfulness applications, including Headspace and Mindfit-Cop [9], contributed to improved emotional regulation and stress management. The RISE program [4] and workplace stress prevention initiatives yielded strong benefits and cost savings, strengthening the business case for investing in mental health programs. Meanwhile, skill-building programs like RESPECT [7] enhanced managerial competency in supporting employee mental health, highlighting the value of leadership-level support for employee mental health.

### 3.4. Variation in Intervention Design

The mental health interventions evaluated across the studies varied widely in duration, structure, content, and implementation strategies, contributing to differences in the economic outcomes. Intervention durations varied broadly, with an average duration of 9.71 months, ranging from 4 h sessions to long-term models spanning up to 9–10 years. Short-term interventions, such as the 4 h face-to-face RESPECT mental health training for managers [7] and the 6-week peer-support messaging program among emergency dispatchers [19], were highly focused and proactive efforts. These programs, by targeting specific behaviors and perceptions over a condensed period, achieved quicker improvements in productivity, reduced absenteeism, and cost savings. Both aimed to enhance social support and management practices in the short term. Conversely, longer term interventions—such as the telemedicine-based collaborative care model for PTSD involving nurse care managers, psychiatrists, and psychologists [16], or the 12-month Mindfit-Cop and Headspace mindfulness app programs [9]—aimed to achieve deeper, sustained improvements in mental health through ongoing support and therapeutic engagement. Programs like Stress-Prevention at Work [15] and the Stepping Stones to Wellness initiative [22], which incorporated multiple components (e.g., e-coaching, biometric screenings, and wellness visits) delivered over six months to a year, aimed for broader improvements in resilience and lifestyle. Another key variation was the intervention’s target level. Programs directed at managerial or supervisory roles, such as RESPECT [7], often had broader organizational impacts by equipping leaders to proactively support employee mental health.

Evaluation methods across the studies included a mix of clinical assessments (e.g., burnout scales, and depression and anxiety measures), administrative data (e.g., absenteeism rates, and turnover rates), and economic models (e.g., Markov simulations, ROI calculations, and cost-effectiveness analysis based on quality-adjusted life years [QALYs]). Remarkably, interventions that were shorter and tightly targeted tended to measure direct workplace outcomes, such as reductions in sick leave and turnover, whereas longer programs often evaluated broader quality of life improvements and clinical recovery metrics.

### 3.5. Cost Variability Across Interventions

The intervention costs varied widely across the studies, depending on the program type and target population. Costs ranged from as little as EUR 50 per employee for a workplace stress reduction program [15] to USD 2029 per patient per year for a telemedicine-based PTSD intervention [16]. Some studies focused on direct training expenses, such as workplace mental health training for managers at AUD 1017 per participant [20], while others included broader implementation expenses—for example, screening 1000 UK police officers for psychological risks cost GBP 84,287 (USD 106,971) [12]. Resilience training and wellness programs fell in the moderate range: a mindfulness-based worksite program [18] cost USD 7662.50 in total, and the PHLAME wellness program for firefighters [10] required USD 600 per participant for team-based interventions and USD 1500 for testing and counseling.

### 3.6. Effectiveness Measures and Outcomes

Studies evaluating these interventions used different approaches to measure effectiveness. Some studies focused on workplace metrics, measuring absenteeism and presenteeism to calculate performance metrics [12], while others, such as [16], used quality-adjusted life years (QALYs) derived from health surveys to determine cost effectiveness. Burnout reduction was another common indicator; for example, ref. [19] measured burnout reduction and tracked turnover and sick leave rates over multiple time points. Several interventions showed significant reductions in sick leave and turnover, with [10] noting no occupational injuries following the implementation of a wellness program.

Most interventions were tailored to specific populations. Muir et al. [20] found nurse retention programs significantly reduced turnover costs, saving hospitals approximately USD 5000 per nurse annually. Peer-led interventions like the PHLAME program [10] proved especially effective in fostering engagement and long-term impact. However, outcome variability remains a challenge, often influenced by factors like organizational context, program design, and the target population.

### 3.7. Financial and Economic Evaluations

Financial outcomes were assessed using metrics such as ROI, cost–benefit ratio (CBR), incremental cost-effectiveness ratio (ICER), and cost-effectiveness ratio (CER), with net monetary benefits (NMB) primarily based on nurse turnover and absenteeism data. For consistency in the meta-analysis, all financial metrics were converted to ROI using standardized assumptions (Appendix A).

After conversion, ROI values across studies ranged from −20% to 3560%. For instance, a surveillance program [12] reported an ROI of 187%, largely due to productivity gains totaling approximately USD 306,713. The PHLAME program [10], aimed at injury prevention and wellness for firefighters, achieved an ROI of 321%, with USD 33,000 in savings from reduced workers’ compensation claims. The highest reported savings came from a peer support program in Germany [5], which yielded an average cost saving of EUR 6672 per participant and a total annual budget impact of EUR 6.67 million. 

### 3.8. Sensitivity Analysis and Variability in Outcomes

Several studies performed sensitivity analyses to validate their findings and to test the robustness of their ROI estimates under varying cost scenarios. For instance, ref. [12] found that ROI estimates ranged from 115% to 296%, while Mindfit-Cop [9] showed a wide cost-effectiveness range from −2136 to −600.57, depending on adjustments in productivity and absenteeism. Overall, while most of the interventions demonstrated clear economic and health benefits, the findings underscore the importance of considering program-specific circumstances and organizational context when evaluating intervention effectiveness.

### 3.9. Assessment of Heterogeneity and Synthesis of Results

Eligible studies for meta-analysis were required to report means along with either standard error or standard deviation values for the outcomes of interest. Both common effects and random effects models were applied to pool effect sizes across the studies using standardized mean differences (SMDs) and their corresponding 95% confidence intervals (CIs).

Heterogeneity was quantified using the *I*^2^ statistics and forest plots [24,25], as shown in Figure 2, to illustrate the variation in the effect sizes across the studies. The analysis revealed high heterogeneity (*I*^2^ = 95.8%), indicating substantial variability among the included studies [26,27]. The pooled effect size was 0.85 (95% CI: 0.74–0.96), suggesting a statistically significant and meaningful overall impact of the interventions. However, the use of a random effects model shows that the effectiveness is not guaranteed in all cases and may be influenced by program design and organizational context. The high level of heterogeneity (*I*^2^) underscores the need for caution when generalizing these results.

Figure 3 presents the analysis of SMDs (Hedges’ g) across various economic evaluation metrics, including ROI, incremental cost-effectiveness ratio (ICER), net monetary benefit (NMB), and cost-effectiveness ratio (CER). A positive SMD or Hedges’ g indicates a favorable economic outcome, while negative values suggest less beneficial results. Interpretation follows Cohen’s criteria [24,25], where an SMD of 0.2 is considered ‘small’, 0.5 ‘medium’, and 0.8 or above ‘large’.

Several interventions demonstrate exceptionally large effects, indicating substantial financial and operational benefits. For instance, the ROI study by McCrone et al. [12] (Hedges’ g = 16.28) stands out with an extremely high ROI value reflecting significant economic gains. Similarly, the occupational physician program (Hedges’ g = 10.42) suggested a high cost effectiveness that confirms its value as a strategic investment [17].

Another important study also exhibited very large effects, emphasizing its substantial economic and health benefits. A 4 h manager training course [7] (Hedges’ g = 8.46) indicated that equipping managers to support employee mental health can result in considerable financial and productivity gains. Similarly, Kuehl et al. [10] (Hedges’ g = 4.53) highlighted the economic advantage of firefighter wellness programs. Ward et al. [23] (Hedges’ *g* = 3.08) and Roesner et al. [5] (Hedges’ g = 4.34) also reported strong effects, reinforcing that workplace mental health interventions can yield significant cost savings and productivity improvements.

While many interventions demonstrated strong economic benefits, some studies indicated more moderate or even negative effects. Muir et al. [20] found a modest financial impact (Hedges’ g = 0.27), while both the Headspace and Mindfit-Cop programs [20] showed negative Hedges’ g values, suggesting these interventions did not generate positive financial returns, likely due to their higher upfront costs. Similarly, sensitivity analyses conducted by Van Dongen et al. [18] reported negative effect sizes (Hedges’ g = −1.85), suggesting that these programs may not be cost effective under specific assumptions. Poor e-coaching compliance may have contributed to this result [18]. As shown in Figure 3, interventions that emphasize early detection and a proactive approach are more likely to have positive economic outcomes. These findings underscore the need to evaluate both direct and indirect financial impacts when assessing intervention effectiveness.

Prevention and stress reduction programs display a high ROI, with prevention programs showing greater variability and stress reduction programs demonstrating more consistent outcomes (see Figure 4). The stability of stress reduction programs indicates their effectiveness in reducing turnover and absenteeism.

Skill-building interventions at the management level reported a high ROI, although this is based on a single estimate, highlighting the need for further research to validate their effectiveness. This aligns with the findings from a prior systematic review, which examined the impact of training workplace managers on employee mental health [28]. It found that such training significantly improved managers’ mental health knowledge (SMD = 0.73), reduced stigmatizing attitudes (SMD = 0.36), and enhanced supportive behaviors toward employees experiencing mental health issues (SMD = 0.59) [28]. These results underscore the potential of manager training programs to increase mental health literacy and promote a more supportive work environment [28].

## 4. Discussion

This meta-analysis provides evidence on the economic and organizational value of mental health and resilience interventions for public safety personnel and frontline healthcare professionals who are frequently exposed to PPTEs [2]. The findings show a clear pattern: interventions designed to be short-term and targeted toward specific occupational stressors are more likely to generate substantial ROI, whereas longer term programs, despite showing clinical benefits, tend to have more variable economic outcomes.

The overall pooled effect size (SMD = 0.85) indicates a statistically significant impact of interventions on economic and operational outcomes, including reductions in absenteeism, turnover, and healthcare costs. However, the high level of heterogeneity (*I*^2^ = 95.8%) underlines the variability across the studies in terms of intervention design, target population, implementation, and outcome measurement, which implies that while resilience and mental health programs are generally beneficial, their success is highly context dependent.

Our analysis indicates that the most cost-effective interventions tend to be proactive, preventive, and delivered over shorter durations (≤6 months). Programs such as the RESPECT training for managers and the peer support initiatives demonstrate immediate productivity gains and reduced absenteeism, contributing to high ROI values. These findings align with the prior literature suggesting that interventions tailored to organizational realities and delivered within operational constraints are more readily adopted and produce measurable impacts in the short term.

Conversely, long-term interventions, such as using telemedicine and mindfulness apps, often involved higher upfront costs and demonstrated delayed or less tangible financial benefits. Under certain assumptions, some of these programs even demonstrated negative ROI values. These outcomes may reflect the inherent challenge of capturing long-term benefits within the constraints of traditional economic evaluation frameworks, which tend to favor short-term, quantifiable returns.

The greater economic effect observed in shorter term programs stems not only from their reduced operational costs but also from their ability to deliver immediate, observable productivity improvements, captured in organizational financial metrics. Meanwhile, the slower developing benefits of longer term programs—such as improved clinical outcomes, quality of life, and reduced future healthcare costs—may be undervalued in short- to medium-term ROI assessments. These findings highlight the need for more comprehensive economic models that better account for the full spectrum of long-term benefits.

A previous study [8] also highlighted that the duration of workplace interventions significantly influences ROI. Shorter programs, such as brief group Cognitive Behavioral Therapy (CBT) sessions, often deliver higher immediate returns by reducing presenteeism. In contrast, longer or more frequent interventions, such as bi-monthly meetings to enhance job control, may result in lower ROI due to increased implementation and employee time costs. However, psychological return-to-work programs involving dedicated therapists tend to maintain positive ROI even over longer durations, primarily because of the substantial reductions in absenteeism [8]. Thus, optimizing the balance between intervention costs and benefits is essential to maximizing economic returns [8].

The level at which an intervention is directed within an organization may influence its overall effectiveness and economic return. Preliminary evidence shows that targeting managerial or leadership roles can yield organization-wide benefits. For example, the 4 h RESPECT managerial training course [7] showed a high ROI value (Hedges’ g = 8.46), indicating that even brief, focused programs that equipped supervisors with mental health support skills can positively affect broader employee well-being. Similar results were also found in a systematic review and meta-analysis [28]. These outcomes suggest that manager training can play a critical role in shaping a psychologically safe work environment [28]. However, similar to our results, the review also indicated that evidence for a direct improvement in employee psychological outcomes remains limited. This highlights the need for future studies to include robust employee-level data to more accurately assess the downstream effects of such interventions on employee mental health.

The variability in cost structures—from EUR 50 per participant to over USD 2000 annually—further reinforces the importance of matching intervention scope and intensity with organizational capacity and workforce needs. Sensitivity analyses suggest that the economic impact of these interventions is sensitive to contextual factors such as program compliance, organizational culture, and workforce engagement. These insights are crucial for policymakers and organizational leaders aiming to invest in scalable, effective mental health strategies.

Although the studies generally reported positive financial and operational outcomes, selection bias remains a concern, as only those studies that reported economic metrics were included. Moreover, some studies lacked standardized outcome measures or failed to disaggregate effects by profession or sector. As a result, our ability to draw precise comparisons across interventions was limited.

Nevertheless, this analysis provides a business case for investing in targeted, well-designed mental health interventions for frontline workers. Future research should aim to standardize economic outcome reporting, incorporate longer term follow-up, and explore sector-specific needs and implementation barriers. Importantly, qualitative assessments of workforce satisfaction and psychological safety should complement economic evaluations to present a holistic view of program impact.

## 5. Limitations

The search strategy had several limitations. While comprehensive and guided by PRISMA guideline, it was restricted to studies that explicitly reported ROI or measured economic benefits.

The limited availability of studies reporting the required statistical measures introduced potential selection bias, which may affect the generalizability and robustness of the meta-analysis findings. This bias also limited the ability to explore the full range of variability across different interventions and populations.

## 6. Conclusions

Findings from the 15 included studies indicate that proactive interventions targeting public safety personnel and frontline healthcare professionals exposed to PPTEs can yield substantial economic and health benefits, particularly when tailored to the specific needs of the target population and organizational context.

ROI varied widely across interventions, from −20% to 3560%. Remarkably, shorter interventions (≤6 months) often yielded higher ROI values, while longer interventions (>12 months) showed more moderate or negative returns. Programs like PHLAME and PSP demonstrated exceptional economic outcomes. Conversely, telemedicine-based PTSD interventions incurred higher costs with limited financial returns, which emphasizes the importance of context-specific considerations for implementation.

An additional insight from this analysis concerns the organizational level at which interventions are implemented. Early evidence indicates that targeting managerial staff may produce broader effects across the workforce. For example, the short-term RESPECT managerial training program demonstrated a notably high ROI, indicating that empowering leaders to support mental health can positively influence workplace culture. Similar findings in [28] also support the effectiveness of leadership-focused mental health programs.

Synthesis analyses further highlighted the variability in intervention effectiveness, which highlights the influence of organizational context, program design, and participant characteristics. While some studies reported large effect sizes (e.g., Hedges’ g = 16.28 for surveillance programs and Hedges’ g = 10.42 for occupational physician programs), others indicated minimal or negative effects, particularly for mindfulness-based digital applications.

In conclusion, workplace mental health interventions are generally cost effective and enhance resilience, reduce absenteeism, and improve retention. However, their success depends on the strategic program design, alignment with organizational priorities, and ongoing evaluation to adapt to changing workforce needs. Future studies should emphasize standardized economic metrics and long-term outcomes to guide evidence-informed decision making. By highlighting key research gaps and practical limitations in existing studies, this review aims to inform the development of high-quality evaluations of program effectiveness, particularly within public safety and frontline healthcare settings. The findings offer valuable insights for organizational leaders and policymakers, presenting a range of opportunities to design targeted mental health strategies that address the specific demands and challenges of these essential frontline roles.

## Figures and Tables

**Figure 1 ijerph-22-00809-f001:**
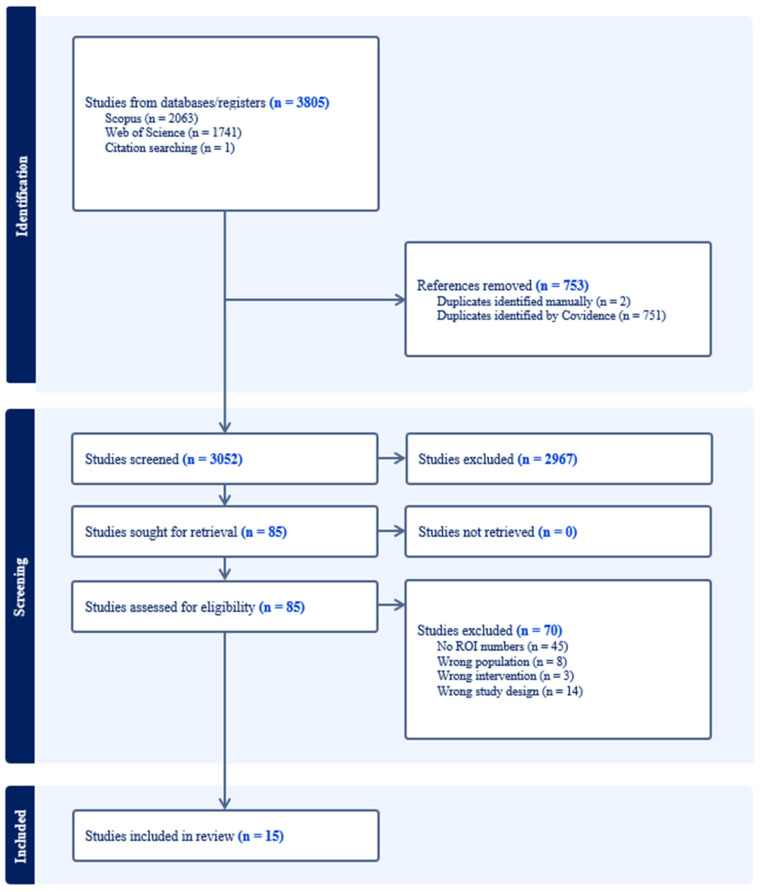
PRISMA diagram.

**Figure 2 ijerph-22-00809-f002:**
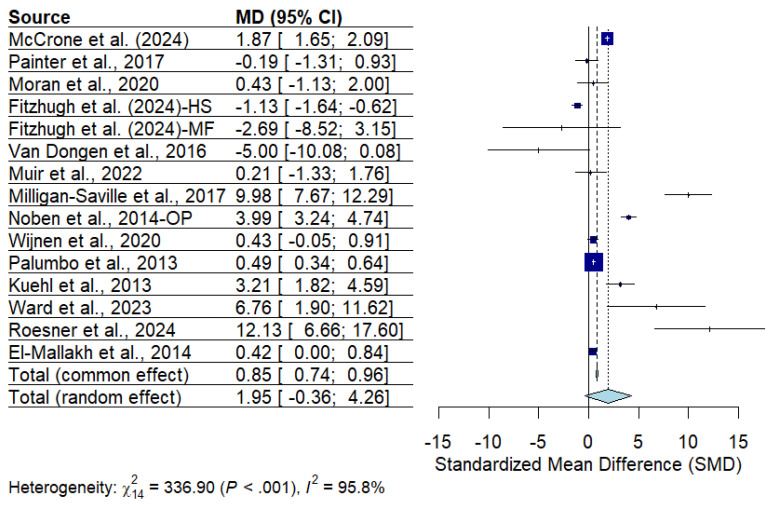
Forest plot [4,5,6,7,9,10,12,15,16,17,18,20,22,23].

**Figure 3 ijerph-22-00809-f003:**
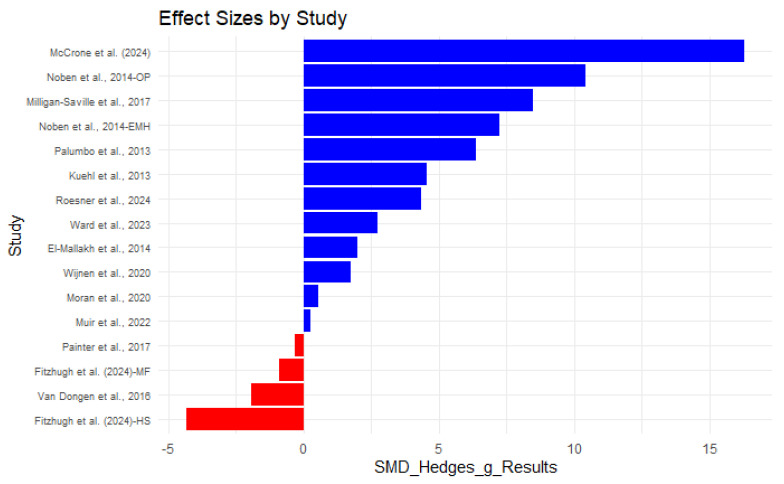
Effect size by study (SMD) [4,5,6,7,9,10,12,15,16,17,18,20,22,23].

**Figure 4 ijerph-22-00809-f004:**
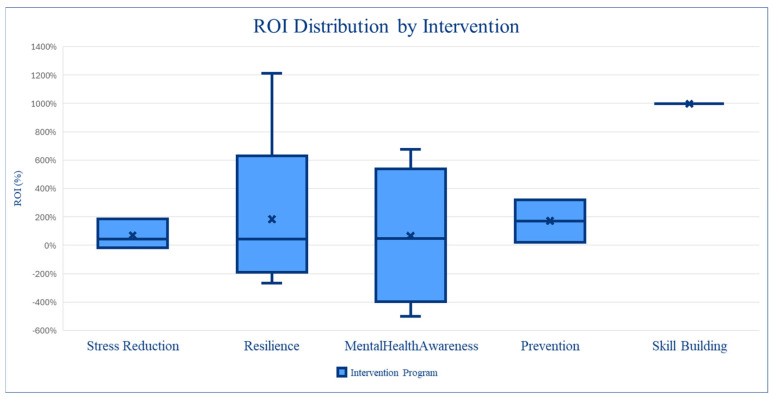
ROI distribution by intervention type.

**Table 1 ijerph-22-00809-t001:** PICO literature search strategy.

Domain	Target	Search Terms
Population	Public safety personnel	Firefighters Police officers Law enforcement Dispatch Communication officers Paramedic Emergency medical technician Emergency medical service First responders Correctional officers Emergency room personnel Nurses Public Safety Personnel
Intervention	Resilience training programs	Prevention Resilience Coping (skills) Family coping Stress reduction Skill building Psychoeducation Mental health awareness (training) Stigma reduction
Comparison	Pre-post interventions	
Outcome	Return on investment	ROI Economic evaluation Cost–benefit

**Table 2 ijerph-22-00809-t002:** Inclusion and exclusion criterion.

Category	Inclusion Criteria	Exclusion Criteria
Population	Adult (≥18 years) public safety personnel and frontline healthcare professionals exposed to PPTEs	Studies involving individuals with clinically diagnosed mental disorders or positive screens on validated psychological instruments
Intervention	Mental health programs designed to proactively mitigate the impact of PPTEs	Studies addressing non-PPTE occupational stressors (e.g., work-related demands, organizational stress)
Outcome	ROI analysis, including pre-post intervention cost assessment	Studies without ROI analysis or cost–benefit evaluation
Language	English-language studies	
Publication Date	Published on or after 1 January 2015	

**Table 3 ijerph-22-00809-t003:** Summary characteristics of eligible studies (n = 15).

Study (Quality)	Sample Size	Population (Country)	Design	Program Description	Program Duration	Evaluation	Key Outcomes	Cost Saving Results
McCrone et al. (2024) [12]	1000	Police personnel (United Kingdom)	Decision analytic model, economic evaluation	Surveillance programtrauma therapy	Over a year	Pre-program, post-program	W/C assessed as an indicator of productivity, factoring in both absenteeism and presenteeism. Monetary measures: productivity gains translated into financial benefits.	The total screening cost was GBP 84,287 (USD 106,971), while the net gain in work productivity was GBP 241,672 (USD 306,713), resulting in a 187% ROI.
Wijnen et al., 2020 [15]	303	Healthcare workers (Netherlands)	RCT	Stress-Prevention at Work intervention for employees	12 months	Pre-program, post-program at baseline, 6, and 12 months.	Productivity losses were measured using the Trimbos and iMTA Cost questionnaire in Psychiatry.	The program delivered significant cost savings for employers, with a modest investment of EUR 50 per employee yielding an average net monetary benefit of EUR 2981 per employee.
Painter et al., 2017 [16]	265	Veterans (United States)	Pragmatic randomized effectiveness	Telemedicine-based collaborative care model for PTSD	12 months	Pre-program, post-program	Effectiveness measured QALYs derived from the Short Form Health Survey for Veterans and QWB scale.	The intervention cost USD 2029 per patient annually and did not lead to healthcare savings, resulting in an incremental cost-effectiveness ratio of USD 185,565 per QALY (IQR: USD 57,675–USD 395,743).
Noben et al., 2014 [17]	617	Nurses (Netherlands)	RCT	Occupational physician condition, e-mental health condition	6 months	Pre-program, post-program-6-month follow-up	Treatment response was defined as an improvement on the Nurses Work Functioning Questionnaire of at least 40% between baseline and follow-up.	At follow-up, WF improved by 20% in the control group, 24% in the occupational physician referral group, and 16% in the e-mental health referral group. The total average annualized costs per nurse were EUR 1752 for the control condition, EUR 1266 for the occupational physician condition, and EUR 1375 for the e-mental health condition. The occupational physician referral was the most cost effective, resulting in cost savings of EUR 5049 per treatment. In contrast, the e-mental health intervention had additional costs, with an incremental cost-effectiveness ratio of EUR 4054.
Van Dongen et al., 2016 [18]	257	Research Institute for Physical Activity, Work and Health, (USA)	RCT	An eight-week mindfulness course that included mindfulness training, e-coaching, and supporting elements.	8 weeks	Pre-program, post-program	Intervention costs, absenteeism, presenteeism, occupational health, healthcare utilization, workplace wellness activities, and work ability are measured.	ROI analysis was performed from the employer’s perspective using Net Benefits (NB), Benefit–Cost Ratio (BCR), and ROI metrics. To quantify precision, 95% bootstrapped CIs were estimated, using 5000 replications.
Linos et al., 2022 [19]	536	911 Dispatchers (United States)	ITT analysis using (RCT)	A 6-week anonymous peer support intervention	6 weeks	Pre-program, post-program	Burnout levels measured via CBI at three points: Baseline, immediately post-intervention and 4 months post-intervention.	Burnout reduction, improved retention, and decreased sick leave led to estimated cost savings of at least USD 400,000 for a mid-sized city.
Muir et al., 2022 [20]	1000	Registered nurses (United States)—hypothetical cohort	Cost–consequence analysis	Implementing a nurse burnout reduction program	10 years	Comparison between 1: status quo 2: burnout reduction program	Burnout prevalence, turnover costs, inclusion/exclusion of burnout programs.	USD 11,592 per nurse per year in the burnout reduction program vs. USD 16,736 per nurse per year in the status quo. Overall savings: USD 5144 per nurse per year from reduced turnover costs.
Milligan-Saville et al., 2017 [7]	128	Fire And Rescue Service (Australia)	RCT	Mental health training program (“RESPECT”) for managers	4 h	Pre-program, post-program-6 month follow-up	Work-related and standard sick leave rates were calculated.	The cost of work-related sickness absence was AUD 10,151.53 per manager less in the intervention group. This ROI is GBP 9.98 for every pound spent on manager mental health training.
Moran et al., 2020 [4]	80	Nursing staff (United States)	Economic evaluation using a Markov model	RISE program as a peer-support program, helps hospital staff cope with stressful events.	1 year	Markov model to compare costs and benefits with and without the program	NMB, the budget impact was calculated based on the program’s cost, nurse turnover, and nurse time off.	The RISE program cost per nurse was USD 656.25. The mean NMB was USD 22,576.05.
Fitzhugh et al. (2024) [9]	477	Police officers and staff (UK)	RCT	Two mindfulness apps aimed at improving well-being: Mindfit-Cop and Headspace	24 weeks, 10 weeks	Pre-program, post-program	(CERs) falling below Layard’s (2016) acceptability threshold of GBP 2500 per additional point [21].	The (CER) was calculated based on improvements in life satisfaction and productivity (measured by absenteeism and performance changes). Both apps were determined to be cost effective, falling below Layard’s (2016) acceptability threshold [21].
Palumbo et al., 2013 [22]	80	USA	Observational evaluation.	Wellness program implemented in a hospital	6 months	Pre-program, post-program	Improved participation and staff satisfaction while potentially reducing absenteeism.	The program led to an estimated cost reduction of USD 11,409.17 in unscheduled absences. With a total intervention cost of USD 7662.50 for 80 employees, the ROI was calculated at USD 3746.67.
Kuehl et al., 2013 [10]	1369	Firefighters (USA)	Retrospective quasi-experimental study.	The PHLAME program, a peer-led workplace health promotion program	12 sessions over the course of the program	Pre-program, post-program	WC claims and medical injury costs were reduced after implementation.	Fire departments participating in the PHLAME TEAM program demonstrated a positive ROI of 4.61–1.00, and the range of ROI is between 1.8 and 4.61.
Ward et al., 2023 [23]	686	USA frontline healthcare service workers	Retrospective cohort study	Digital mental health benefit	6 months	Pre-program, post-program	Clinical improvement measures were PHQ-9 scale for depression and GAD-7 scale for anxiety; workplace measures were employee retention and SDS for functional impairment.	Participants reported 0.70 (95% CI, 0.26–1.14) fewer workdays per week impacted by mental health issues, corresponding to USD 3491 (95% CI, USD 1305 USD 5677) salary savings at approximately federal median wage (USD 50,000).
Roesner et al., 2024 [5]	1000	nursing staff—German	Economic evaluation using a Markov model	Peer support program aimed at reducing the psychosocial burden from the second victim phenomenon.	1 year	Pre-program, post-program	Reduced sick days, turnover costs, and absenteeism.	Average cost saving of EUR 6672 per healthcare worker. Main reason for the reduction is the reduction in dropouts, whereas the costs of sick day leaves are only moderately affected. The expected annual budgetary impact is estimated to be approximately EUR 6.67 M in the hospital considered.
El-Mallakh et al., 2014 [6]	1454	Police officer—USA	Cost-Effectiveness Analysis (CEA)	(CIT) program	9 years	Pre-program, post-program	Cost savings are due to reduced hospitalization and jail time.	Acost savings analysis of a CIT program in a medium-size southern city was performed. The annual costs of a CIT program were USD 2,430,128 and the annual savings were USD 3,455,025.

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
