# Peer review of "Economic Evaluation of Proactive PTSI Mitigation Programs for Public Safety Personnel and Frontline Healthcare Professionals: A Systematic Review and Meta-Analysis"

_ijerph, 2025, doi:10.3390/ijerph22050809_

Round 1

Reviewer 1 Report

Comments and Suggestions for Authors

The review methods and results are clear and well communicated.

The introduction and discussion/conclusion are much too brief and need to acknowledge that the effectiveness of PTSI prevention programs is in doubt to begin with. A single citation in the introduction is inadequate to acknowledge research to date on PTSI prevention, there is no discussion section, and the conclusion is very brief and has no citations.

I am very surprised not to see this key article cited in any way, which found "The effectiveness of various organizational programs designed to “prevent”—or more accurately to proactively mitigate—PTSI and improve psychological health indicators among PPTE-exposed occupational groups remains unclear":

Di Nota, P. M., Bahji, A., Groll, D., Carleton, R. N., & Anderson, G. S. (2021). Proactive psychological programs designed to mitigate posttraumatic stress injuries among at-risk workers: a systematic review and meta-analysis. Systematic reviews, 10(1), 126.

Author Response

  1. Thank you for the helpful feedback. We have expanded the introduction and added dedicated discussion sections. The discussion and conclusion now address key points, including the economic impact of PTSI, the role of organizational and managerial-level interventions, and important considerations for program design. Relevant literature has been cited to strengthen these sections and acknowledge the complexity of PTSI prevention efforts.
  2. The key article was an oversight. Thank you for pointing this out. We have now cited Di Nota et al. (2021) in the introduction. 

Reviewer 2 Report

Comments and Suggestions for Authors

Dear authors, thank you for the opportunity to review the results of your meta-analysis.
In their work, the authors conduct an economic evaluation of programs for proactively reducing the risk of PTSI for public safety personnel and first-line health workers. The authors describe in detail the technology for selecting articles for analysis, the criteria for their selection and subsequent analysis, and clearly present both the stages of the study and the selected articles in the form of tables. The results of the review are described qualitatively, which allows us to conclude that the findings are valid. This study contributes to the development of occupational hygiene and medicine in terms of preventing and correcting PTSD in two vulnerable professional groups: public safety employees and health workers, and emphasizes the importance and economic feasibility of these programs.

At the same time, there are a number of recommendations for the authors to improve the manuscript:
1. Please supplement the manuscript with more content of the programs themselves, which showed different economic effects. The authors mentioned that shorter-term programs gave a greater economic effect. But I would like to know what the content of these programs is, perhaps the reason for the better effect lies in the methods of influence themselves. 2. It is possible to supplement the results discussion section, where it is necessary to mention studies that were not included in the 15 selected, but claimed high-quality results, but the data on effectiveness were not confirmed. This would allow to describe the directions of future research, which are not clearly outlined now.
3. It would be nice to see a more extensive review of sources in the introduction and discussion of results, since the list of sources 24 is too small for a review study. Expanding the sources of literature is necessary for a high-quality description of existing gaps in research and future directions of research.
The recommendations made do not detract from the overall positive impression of the work.
The manuscript requires minor changes and additions.
Best wishes, reviewer.

Author Response

  1. Thank you for your suggestion. In response, we have added Section 3.4 to provide more detail on the content of the programs, highlighting key components such as delivery methods and areas of focus that may help explain differences in outcomes beyond program duration. We have also expanded the Discussion (paragraphs 3–5) to explore potential underlying reasons for the observed variation in economic effects. 
  2. We have expanded the discussion to address key points, including the economic impact of PTSI, the role of organizational and managerial-level interventions, and important considerations for program design. The revised section also includes a discussion on directions for future research. We did not expand the papers included in the table to stay true to our methods
  3. As outlined in our abstract screening process, we initially reviewed approximately 3,052 papers, of which only 85 met our inclusion criteria based on the PICO framework for PSP and FHP exposed to PPTEs. Of these, only 15 reported economic metrics relevant to our analysis, highlighting a significant gap in the literature on the economic evaluation of PTSI interventions in high-risk occupations. Identifying this gap has been a critical part of our learning and reflects the importance of our inclusion parameters in shaping the scope of the research. 

    We recognize that the current list of 24 references may seem limited for a review study. In response, we have expanded the introduction and discussion sections to include a broader range of literature, providing a more comprehensive view of existing research gaps and future directions. 

Reviewer 3 Report

Comments and Suggestions for Authors

Abstract
The abstract does not clearly present the objectives of the study.
It does not report the main findings. It is also necessary to include the key conclusions of the study.

Introduction
It is important to provide a conceptual clarification of the central topic: “posttraumatic stress injuries (PTSI)”.
What are posttraumatic stress injuries (PTSI)? What are potentially psychologically traumatic events (PPTEs)?
What is the impact of PTSI on public safety personnel and frontline healthcare professionals? What studies already exist on this topic? Why is this study relevant?

Methods
The exclusion criterion based on “collectivist” countries lacks methodological justification and may be considered biased.
Was the population composed of “veterans”, “police officers”, “nurses”, and “health professionals”? Why were these very distinct professional groups chosen as the target population?
How was the quality of the evidence assessed? Was the GRADE approach used?

Results
It would be important to include a table summarizing the main results.

Conclusions
The conclusions are too general and do not provide concrete recommendations for the implementation of programs.
It is suggested to highlight: The most effective types of programs,The optimal duration of interventions, and Implications for occupational mental health policy.

Comments on the Quality of English Language

It is recommended to revise long and poorly structured sentences by breaking them into shorter, more cohesive statements.
Repetitions and redundancies should be eliminated to improve clarity and conciseness.
A professional language review—ideally conducted by a native speaker or scientific editor—is strongly advised.
Technical terms and abbreviations should be standardized throughout the manuscript.
Clear transitions between paragraphs and ideas (e.g., "Therefore", "Furthermore", "However") should be used to enhance the flow of the text.

Author Response

  1. We appreciate your suggestion. In response, we have revised the abstract to clearly outline the study's objectives, key findings, and main conclusions.   
  2. Both  "posttraumatic stress injuries (PTSI)" and "potentially psychologically traumatic events (PPTEs)" have been described in more detail to be certain readers understand the intended use of the terms such as in the revised first paragraph of the introduction. 
  3. The impact of PTSI on public safety personnel and frontline healthcare professionals has been added to lines 36-38.
  4. We have added a justification for the exclusion of studies from collectivist countries in Section 2.3 (lines 150–157), explaining the rationale based on differences in workplace structures, mental health service access, and cultural factors that may affect generalizability to our target context. 
  5. We have addressed this point in Section 2.3 (lines 144–149), explaining that these professional groups were included due to their shared exposure to PPTEs and similar occupational risks for developing PTSIs. 

    As the study design was part of the inclusion criteria we did not use a quality of evidence scheme such as GRADE as scores would be clustered.

  6. Thank you for your suggestion. We have visually presented the main results using a forest plot, a program interventions graph, and a PRISMA diagram outlining our methodology. These visual tools summarize key findings and study selection in a clear and accessible format.  
  7. Thank you for your comment. We have revised the conclusion to address your suggestions by adding specific points. These are now reflected under the themes of Economic Impact, Organizational Factors (managerial-level interventions), Program Design Considerations, and Future Recommendations.
  8. Thank you for your comment. We have revised the conclusion to address your suggestions by adding specific points. These are now reflected under the themes of Economic Impact, Organizational Factors (managerial-level interventions), Program Design Considerations, and Future Recommendations.

Round 2

Reviewer 1 Report

Comments and Suggestions for Authors

Thank you for your significant revisions to this manuscript. The findings are now situated in the broader literature, and the applicability of the research to different contexts is critically considered. I have no further revisions to suggest.

Reviewer 3 Report

Comments and Suggestions for Authors

In my assessment, the revisions proposed are well-justified and have effectively addressed the key issues raised, making the manuscript suitable for publication.